# Adaptation and validation of the Weight Efficacy Lifestyle Questionnaire (WEL) in a Chilean sample

**Mariela Gatica-Saavedra**[1], **Gabriela Nazar**[2], **Patricia Rubí**[1], **Claudio Bustos**[2]*

**1** Department of Psychiatry and Mental Health, Faculty of Medicine, Universidad de Concepción, Concepción, Chile, **2** Department of Psychology, Faculty of Social Sciences, Universidad de Concepción, Concepción, Chile

* clbustos@udec.cl

**Data Availability Statement:** All data files are available from the OSF database: https://osf.io/s9uzf/?view_only=3380828e78e44cc790c12714faeb00a4.

## Abstract

Self-efficacy is a cognitive-emotional factor that is consistently associated with behavioral change and, in particular, with changes in health behavior. Eating self-efficacy, understood as adopting and maintaining behaviors such as controlling one's weight and trusting in one's ability to control one's eating behavior, has been proposed for managing obesity. This study aimed to validate the Chilean version of the Weight Efficacy Lifestyle Questionnaire (WEL) in a sample of adults from the general population. Four hundred sixty-nine individuals (69.08% women, mean age = 38.02; SD = 10.31) participated in the study. An instrumental design was used geared toward developing tests and psychometric instruments, including adapting existing ones. Exploratory and confirmatory factor analyses were performed. The instrument version validated in Spain was applied in the study. The analysis obtained an instrument of 11 items with adequate psychometric properties allowing its use in clinical and research settings. It can help assess eating self-efficacy in the general population.

## 1. Introduction

Self-efficacy is a cognitive-emotional factor consistently associated with behavioral change, particularly in health behaviors. It is understood as the individual's confidence in their ability to engage in a particular conduct in the face of apparent difficulties or challenges [1]. It is related to how they feel about themself and how this influences their actions, the effort they exert to do so, and how persistent they are in achieving a goal [2].

Self-efficacy can be understood as a generally held ability [1] and a specific resource within a particular domain. It has thus been described as having a positive impact on self-health care [3], and its influence is recognized on such diverse behaviors as healthy food intake [4], regular exercise [5], and controlling one's alcohol and tobacco consumption [6] to cite a few examples.

Applying the concept of eating self-efficacy to obesity management has been proposed, understood as adopting and maintaining weight control behaviors and trusting in one's ability to control personal eating behavior [7].

**Funding:** The primary author is a grantee of the National Doctoral Program of the Agencia Nacional de Investigación y Desarrollo (National Agency for Research and Development - ANID), Chile. The funders had no role in study design, data collection and analysis, decision to publish, or preparation of the manuscript.

**Competing interests:** The authors have declared that no competing interests exist.

Various studies have evaluated the relationship between eating self-efficacy and eating behavior, finding associations with lower caloric intake in free-choice meals [8], less emotional, uninhibited eating [9], low consumption of high-calorie snacks [10], improvement in behavioral treatments aimed at weight loss [11, 12], improvement in the preference for consuming fruits and vegetables rather than fast food, mediated by social support [13].

## 1.1 The Weight Efficacy Lifestyle Questionnaire (WEL)

The WEL is a self-reporting instrument developed to evaluate the role of eating self-efficacy in treatment obesity [14]. It consists of 20 items referring to different situations in which the respondent grades their confidence in their ability to resist the desire to eat. The items are organized in five domains: (1) negative emotions (e.g., "I can resist eating when I am depressed [or discouraged]"); (2) availability (e.g., "I resist eating even when high-calorie foods are available"); (3) social pressure (e.g., "I resist eating even when I have to say ´´´no´´´ to others"); (4) physical discomfort (e.g., "I can resist eating when I am in pain"); and (5) positive activities (e.g., "I can resist eating when I am happy"). Each item is answered on a Likert scale from 0 (I am not at all confident) to 10 (I am totally confident). Each domain scores points from 0 to 36 for an overall score of 0 to 180. The WEL can be interpreted as a global score or as five different subscales and is often used as a screening instrument for identifying the resources and limitations of individuals undergoing weight loss treatment. The background information on the WEL indicates that it has adequate psychometric properties. In the authors' validation of the instrument using a sample of patients with obesity [14], internal consistency showed Cronbach's α values of 0.7 to 0.9 in the different subscales, which are all acceptable [14]. The same authors later performed cross-validation with a new sample and obtained similar results. To evaluate criterion validity, the Eating Self-Efficacy Scale (ESES) [15] was used, a 25-item instrument that was developed based on Bandura's Self-Efficacy Theory (1977) [1]. The ESES is used to measure self-efficacy about dietary intake. Correlation analyses indicated a significant association with the WEL, with scores that ranged from r = -0.55 to -0.67. The scale consists of 25 items and two factors: Negative Affect (NA) and Socially Acceptable Circumstances (SAC). The ESES demonstrated good internal consistency, test-retest reliability, and convergent validity [15].The WEL has been validated in the Spanish population [16]. The results did not support the five-subscale structure initially proposed by Clark et al. [14], as only three were confirmed: negative emotions, social pressure, and physical discomfort. Notwithstanding this, the results were sound, as shown by the high factorial weight of the respective factors analyzed corresponding to each of the three groups of participants (range between 0.7 and 0.92). To calculate reliability, Cronbach's α was used for the complete test and the subscales. Concerning the total WEL scores, α values of 0.91, 0.85, and 0.88 were obtained for the clinical population, overweight, and normal weight groups, respectively. As for the subscale of Negative Emotions, the scores obtained by the three groups were 0.94, 0.77, and 0.85, respectively. The Social Pressure subscale scores for the three groups were 0.89, 0.79, and 0.82, respectively. Finally, the scores of the three groups in the subscale of Physical Discomfort were 0.83, 0.78, and 0.68, respectively. A problem was detected with the translation and cross-translation of the original scale, specifically Item 19, the term "uncomfortable" in the Spanish validation, in "I can resist eating when I feel uncomfortable" ("Yo puedo resistir comer cuando me siento incómodo"). The issue was that the semantic content of "uncomfortable" in English refers to a physical nuance, whereas the Spanish translation "incómodo", refers to psychological as well as physical discomfort [16]. The differences between the English version and the Spanish adaptation have been explained, citing cultural differences. For example, some of the items might have little relevance in the Spanish cultural context. This is the case of Item 12: "I can resist eating even

when I am at a party". There are most likely other social events in Spanish culture where food is always on hand (e.g., first communions, baptisms, and weddings); however, this is not the case with "fiestas" or parties, where drinks and music play a much more important role than food, which may or may not be served [16]. This example is consistent with Chilean culture and its legacy of Spanish customs, resulting in Chilean social practices being more similar to Spanish ones than those in the English version of the scale. However, it is important to adapt the instrument to the target population. In the French validation of the instrument [17], the authors proposed a general population questionnaire with 12 items and another one for a clinical population comprising 11 items. Two factors were obtained in both cases: "Internal stimulus" and "External stimulus". The French version eliminated the same items that the Spanish version left out of the original WEL in English, while also removing the following item that the Spanish version has retained: "I am able to resist eating even if I have a headache". And the French version retained the item: "I am able to resist eating even when I am at a party", which the Spanish version removed.

Recent studies have found that the WEL is a robust instrument for assessing an individual's efficiency in incorporating behavior that can support weight loss. The successful attempts of participants to lose weight have been found to correlate with an overall improvement in their WEL scores, both before and after treatment, which is an indicator of adequate external validity [18].

Given the importance of self-efficacy for weight control and the evidence *vis-à-vis* its role in managing obesity, it is vital to have ready access to valid and reliable instruments in the local context. Therefore, this study proposed to validate the WEL in a sample of Chilean adults from the general population.

## 2. Material and methods

An instrumental design was used that was geared to the development of psychometric tests and instruments, including their adaptation [19]. The Spanish version of the instrument was used in the validation [16].

### 2.1 Participants

Westland [20] was used as a reference for calculating the sample size. Therefore, a sample of 200 people was required to model the original structure of 12 items in 3 factors. The minimal sample was thought to be 400 people, which an exploratory factor analysis would be used to verify.

The participants numbered 469 individuals from Chile's general population. Of this total, 324 (69.08%) were women, 138 (29.42%) men, and two participants (0.43%) identified as non-binary. Their ages varied from 18 to 75 years (M = 38.02; SD = 10.31). The body mass index (BMI) showed a mean of 26.12 (SD = 4.43), corresponding to overweight nutritional status. Of the total number of participants, 11 (2.3%) were in the undernourished category, 197 (42%) in the normal category, 186 (40%) were in the overweight range, 55 in the Type I obesity range (12%), 17 in the Type II obesity range (3.6%), and three in the range of morbid obesity (0.6%).

The participants were asked if they had tried to lose weight. Four hundred sixty-two people answered, of whom 401 had tried (86.79%), and 61 (13.2%) had not tried. Regarding the question "Have you been diagnosed with obesity", 459 participants responded, of whom 320 indicated that they had been diagnosed with it (69.72%).

The question, "If you have tried to lose weight, have you succeeded?" was answered by 399 participants, and the response options were: Yes, what I expected (152 people, 38.09%); Yes,

almost what I expected (107 participants, 26.81%); Yes, but very little, less than what I expected (121 participants, 30.32%); and No, I did not manage to lose weight (19 participants, 4.76%).

## 2.2 Procedure

The participants were recruited through open calls via networking, informal groups, and social media. Those interested in participating were given a link to a site with information on the study's features and were asked to sign an informed consent form.

Prior to the start of the study, a cognitive interview [21] was conducted with three people selected based on convenience sampling. They were asked to answer the questionnaire, after which they were queried on their understanding of the items, social desirability, sensitivity, and the adequacy of the multiple-choice answers. No suggestions for changes were made.

The instrument was the version of the WEL validated in Spain [16], comprising 12 items and three subscales (Negative Emotions, Social Pressure, and Physical Discomfort). The response options ranged from 0 to 10, where 0 indicates that the respondent considers they cannot cope with the situation described, and 10 indicates that they feel completely confident of being able to do so.

The participants also answered the Spanish version of the ESES [15, 22]. This instrument consists of 25 items that measure two hypothetical dimensions of self-efficacy in managing one's diet: Negative Affect and Socially Acceptable Circumstances. The responses are presented on a Likert response scale from 1 to 7, with responses 1 and 2 corresponding to "I would have no difficulty controlling myself", and responses 3 to 5 range to "I would have moderate difficulty", and 6 and 7 to "It would be very difficult to control myself". The Spanish validation supported the two-factors structure and showed adequate internal consistency (0.92 in normal-weight participants and 0.93 in the overweight group).

In addition, a survey was administered on the sociodemographic background (age, sex) and a health history questionnaire that included questions on weight, height, obesity diagnosis, and health conditions that possibly influenced body weight (e.g., cancer, surgeries); current treatments to lose weight; a diagnosis of diabetes or insulin resistance, and whether the individual was undergoing treatment; minimum and maximum weight in adulthood; and a diagnosis of anorexia or bulimia.

Concurrent validity was additionally determined through the following questions: First, "If you have tried to lose weight before, have you succeeded?". The response options were: "Yes, I lost the weight I expected to"; "Yes, I lost almost what I expected to"; "Yes, but I lost very little, less than I expected"; and "No, I was unable to lose weight".Another question was: "To what extent do you feel capable of controlling what you eat in order to lose weight if you set yourself this goal?" The response options ranged from "Very capable" 5 to "Not at all capable" (1).

The R program performed the analyses with complete data only. For this reason, no imputation or missing data techniques were used.

The instrument was hosted on the LimeSurvey platform, and the data were collected from October 2019 to January 2020.

This study was approved by the Faculty of Medicine's Scientific Ethics Committee at the Universidad de Concepción (CEC Code 36/2019). The individuals contacted were invited to participate and were informed of the study objectives and how much time would be needed to answer the questionnaires. No incentives were offered for participation.

## 2.3 Data analysis

The data analysis strategy considered the following: a) exploratory and confirmatory factor analyses, b) reliability analysis of the instrument, and c) criterion validity analysis.

The total sample was randomly divided in two to perform an exploratory factor analysis (EFA) in one subsample and a confirmatory factor analysis (CFA) in the other [23].

For the EFA, descriptive analysis was first performed to assess item appropriateness. The Kaiser-Meyer-Olkin statistic and Bartlett's test of sphericity were calculated to define the relevant criteria for the analysis. Horn's parallel analysis, Velicer's Very Simple Structure (VSS), Minimum Average Partial (MAP), and Bayesian information criterion (BIC) analysis were used to determine the number of factors. Least squares and oblique rotation (Oblimin) were used as the factor extraction method. An iterative procedure was established to omit items with a factor loading of less than 0.4 or cross-loadings. We opted for a minimum of 3 items per factor [24].

As for the CFA, the model obtained through the EFA was tested on the second half of the sample using the lavaan package [25]. Considering that the response scale of the items is 0 to 10, the robust maximum likelihood method (ML) was considered the estimator. The model's goodness of fit was analyzed using the following indices: $\chi2/df$, comparative fit index (CFI), Tucker-Lewis index (TLI), root mean square error (RMSEA), and standardized root mean square (SRMR). An $\chi2/df$ fit was considered an adequate fit when $X^2$ did not exceed three times the degrees of freedom (df), when the CFI and TLI values were higher than 0.95, and when the RMSEA and SRMR values were lower than 0.8, considering the values lower than 0.6 and 0.5, respectively, as excellent [26].

For the reliability analysis of the instrument, the homogeneity of the items, Cronbach's Alpha and the Composite Reliability Index were analyzed for each subscale.

To analyze criterion validity, a correlation analysis was performed between the variables of interest (BMI and the questions: Have you ever tried to lose weight?, and If you have tried to lose weight, have you succeeded?) and the ESES. All of the statistical analyses were performed with the R software, version 4.

# 3. Results

## 3.1 Item description

An exploratory, descriptive analysis was performed to evaluate the adequacy of the items (n = 239). The means are slightly higher than the theoretical mean (5.5). Regarding skewness and kurtosis, none of the items presents absolute values greater than 1 (See Table 1).

**Table 1. Descriptive subsample exploratory factor analysis (n = 239).**

| Variable | Mean | SD | Skew | Kurtosis |
|---|---|---|---|---|
| B1 | 5,93 | 2.64 | -0.17 | -0.97 |
| B2 | 6.77 | 2.64 | -0.48 | -0.77 |
| B3 | 7.52 | 2.74 | -0.94 | -0.31 |
| B4 | 6.63 | 2.93 | -0.46 | -1.10 |
| B5 | 7.42 | 2.63 | -0.84 | -0.25 |
| B6 | 7.49 | 2.74 | -0.92 | -0.21 |
| B7 | 7.35 | 2.77 | -0.81 | -0.57 |
| B8 | 7.19 | 2.57 | -0.65 | -0.61 |
| B9 | 7.95 | 2.42 | -1.06 | 0.07 |
| B10 | 6.68 | 2.87 | -0.48 | -1.04 |
| B11 | 7.29 | 2.55 | -0.74 | -0.39 |
| B12 | 7.22 | 2.76 | -0.72 | -0.61 |

### 3.2 Factor structure adequacy

The Kaiser-Meyer-Olkin (KMO) Index and Bartlett's Test of Sphericity confirmed that the correlation matrix was suitable and the data adequate for factor analysis (KMO = 0.94; X2 (66) = 3546.585, p<0.001). Through the application of Very Simple Structure (VSS) analysis, two factors were identified, Velicer's MAP obtaining a minimum of 0.03 with 1 factor and BIC obtaining a minimum of 4 factors. Horn's parallel analysis was used to determine the number of factors, and it identified one. The decision was made, therefore, to test from 1 to 3 factors.

The least squares extraction method was used, subjected to Oblimin rotation. The various models were tested with one, two, and three factors. The one factor model presented adequate loadings on all the items (between 0.69 and 0.85) and an explained variance of 60%. The two factor model explained 65% of the variance, and the items were clearly divided into two factors, with loadings ranging between 0.9 and 0.54 and 1.00 and 0.57, and the three factor model explained 69% of the variance, but no items load on factor 3. Therefore, considering that previous studies had identified two factors, it was decided that the two-factor solution would be maintained.

Considering the content of each subscale, the names "Physical and emotional discomfort" and "External pressure" were assigned. The first factor, designated as "Physical and emotional discomfort", includes 7 items (initial codes: b9, b7, b3, b6, b12, b10, and b4). It assesses the difficulty of refraining from eating when the subject is in negative valence emotional states related to mood and physical discomfort.

The second factor, termed "External pressure", includes 5 items (initial codes: b2, b1, b11, b5, and b8). This factor evaluates how difficult it is to resist eating under environmental pressure. It also includes an item that measures anxiety as a factor of the pressure of overeating.

### 3.3 Confirmatory analysis

A CFA was performed to test the one- and two-factor structure against the second half of the sample (n = 223).

The indicators goodness of fit for the one-factor model were as follows: $X^2(54)$ = 186.231, p<0.001; X2/df = 3.44; CFI = 0.851; TLI = 0.818; RMSEA = 0.105; SRMR = 0.074. The CFI and TLI were lower than expected. The RMSEA was well above the expected value, as was the SRMR, which was found to be above 0.05 but still within an acceptable range (up to 0.8) [26].

Regarding the indicators goodness of fit for the two-factor model, the results were $X^2(53)$ = 107,072, p<0,001; $X^2$/df = 2,02; CFI = 0.939; TLI = 0.924; RMSEA = 0.068; SRMR = 0.053. The CFI and TLI indices were slightly lower than expected in the two-factor model. At the same time, the RMSEA presented a sufficient fit, and the SRMR was higher than expected, although still acceptable [26].

Upon reviewing the modification index, there was a negative relationship between Items b4 and b9 that was slightly higher than expected (21.61). The remaining items appeared adequate. Due to the above, it was decided that a new confirmatory analysis would be performed without including Item b4 in Factor 1. The results were X2(43) = 79.961, p<0.001; X2/df = 1.86; CFI = 0.952; TLI = 0.939; RMSEA = 0.062; SRMR = 0.047. The results improved, but it was decided to test the fit without item b9 in factor 1, but keeping factor b4 due to its theoretical value (I can resist eating when I am depressed (or down)). The results were X2(43) = 77.33, p<0.001; X2/df = 1.798; CFI = 0.956; TLI = 0.943; RMSEA = 0.060; SRMR = 0.054. Since the best fit was achieved without item b9, we kept item b4 and eliminated item b9. Table 2 presents the factors, items, and respective loadings.

**Table 2. Weight Efficacy Lifestyle Questionnaire: Factors, items, and loading.**

| Factor and item | | Factor loading |
|---|---|---|
| **Physical and emotional discomfort** | | |
| 12 | I can resist eating when I feel uncomfortable. | 0.91 |
| 4 | I can resist eating when I am depressed (or down). | 0.80 |
| 3 | I can resist eating when I feel physically run down | 0.79 |
| 7 | I can resist eating when I am angry (or irritable). | 0.78 |
| 10 | I can resist eating when I have experienced failure. | 0.78 |
| 6 | I can resist eating even when I have a headache. | 0.59 |
| **External pressure** | | |
| 2 | I can resist eating even when I have to say "no" to others. | 0.86 |
| 11 | I can resist eating even when I think others will be upset if I don't eat. | 0.85 |
| 5 | I can resist eating even when I feel it's impolite to refuse a second helping. | 0.85 |
| 8 | I can resist eating even when others are pressuring me to eat. | 0.70 |
| 1 | I can resist eating when I am anxious (nervous). | 0.50 |

## 3.4 Reliability analysis

The internal consistency of the scales was calculated using Cronbach's α, and both presented adequate levels. The "Physical and emotional discomfort " subscale presented α = 0.9, and the "External pressure" subscale, α = 0.89. The corrected item-total correlations were also acceptable in all of the cases. The range for the subscale, "Physical and emotional discomfort", was 0.63 to 0.79, and for the "External pressure" scale, it was 0.71 to 0.85, which indicated that all of the items contributed to the scale's internal consistency.

The composite reliability calculation was also performed. The "Physical and Emotional Discomfort" subscale presented a value of 0.902, and the "External Pressure" subscale presented a value of 0.872.

## 3.5 Criterion validity

Criterion-related validity was determined by examining the empirical relationships between the self-efficacy scores and other theoretically-related variables: (a) BMI and (b) prior attempts to lose weight and the outcome of these attempts.

The expectation was that the WEL scores would be negatively related to BMI and prior attempts to lose weight and their outcomes.

However, no significant relationship was found after assessing the relationship between each factor and the question: "Have you tried to lose weight?". Concerning "Physical and emotional discomfort", the factor presented t = 1.3372, df = 75.184, p = 0.1852, while the values for the factor "External pressure" were t = 1.4044, df = 75.426, p = 0.16.

Regarding the relationship between each factor and the question "If you have tried to lose weight, have you succeeded?", significant values were found with a result of r = 0.27 for the "Physical and emotional discomfort" subscale and an r = 0.31 for the subscale "External pressure". Figs 1 and 2 show that higher levels of self-efficacy were presented by those who successfully reduced their body weight.

With respect to the analysis of the relationship between each factor and BMI (see Figs 3 and 4), a third-order polynomial relationship was observed in which two inflection points were present. The self-efficacy levels for both factors are lower at BMI values below 20, then rise progressively until reaching a maximum BMI of 20. They then progressively decrease until reaching a BMI of 35, after which they rise again to maximum levels. Both factors exhibit the same pattern when compared to BMI.

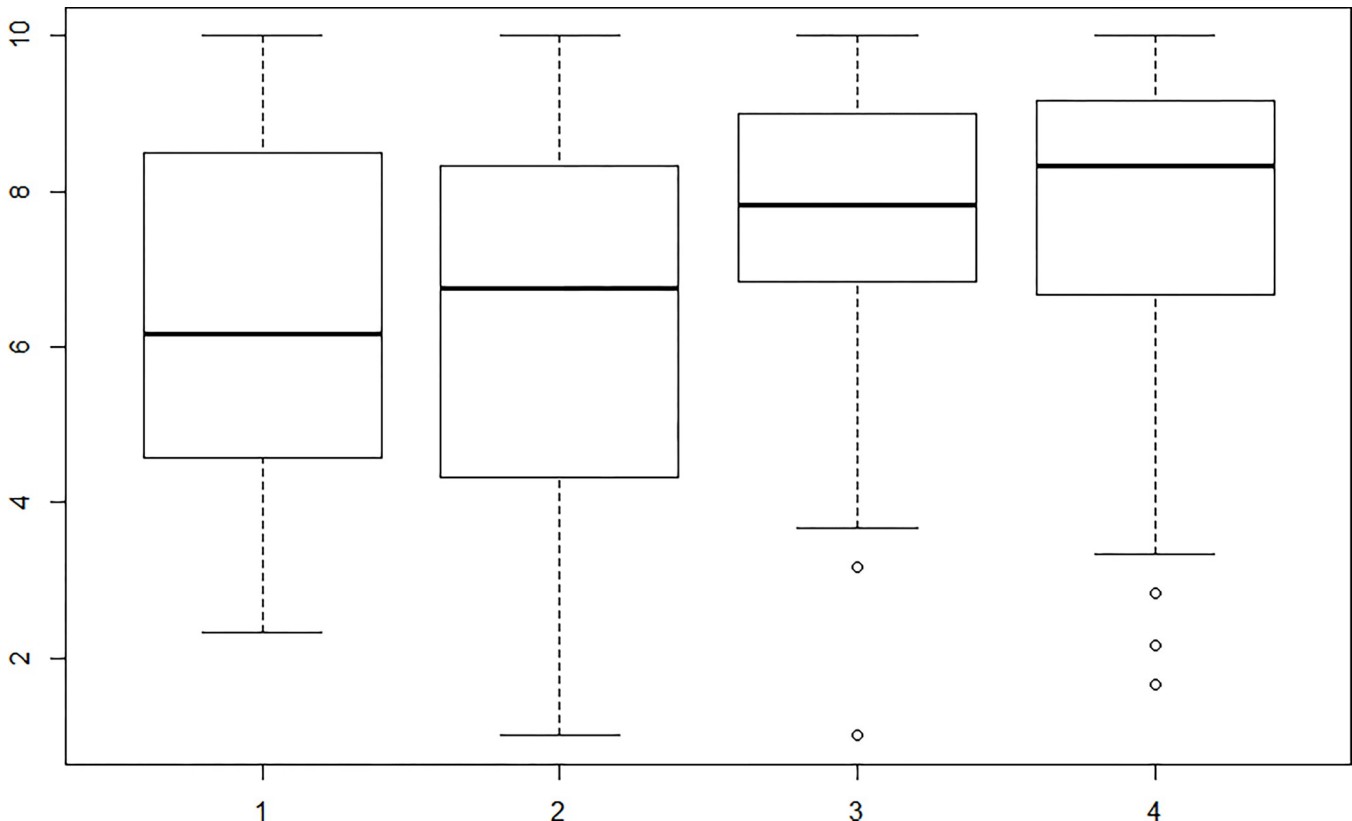

**Fig 1. Relationship factor "Physical and emotional discomfort" and the question "If you have tried to lose weight, have you succeeded?".** The scores are equal to the following answers: 1. No, I did not lose weight; 2. Yes, but very little, less than what I expected; 3. Yes, almost what I expected; 4. Yes, what I expected.

An analysis was also performed that compared BMI to each factor, separating those who had tried to lose weight from those who had not. With respect to the factor "Physical and emotional discomfort", for participants who had indeed tried to lose weight, those with a BMI less than 20 had very low self-efficacy scores, which rose to a BMI of around 22, reached a maximum BMI of 30, which then began to decline. As for patients who had not tried to lose weight, those with a BMI of less than 20 started with high scores that declined progressively until a BMI of 35 was reached, from which they rose again, back to high levels. A similar trend is observed concerning the "External pressure" factor. Participants who had tried to lose weight had a low self-efficacy score at BMIs below 20, the highest point reached, after which it began to decrease (Figs 5 and 6).

The correlations between WEL and ESES and among their respective subscales with similar theoretical content were calculated. The correlation coefficient between the two scales was r = -0.65. The subscales "Physical and emotional discomfort" of the WEL and "Negative affect" of the ESES presented r = -0.63. The "External pressure" scale of WEL and the "Socially acceptable circumstances" scale of ESES showed r = -0.51. The negative sign is because high ESES scores indicate low self-efficacy, while the opposite occurs in the WEL; therefore, these associations trended as expected.

## 4. Discussion

The WEL was developed and validated to evaluate the role of eating self-efficacy in treating obesity [14]. This study used the version previously validated in the Spanish population in a sample of the general Chilean population [16].

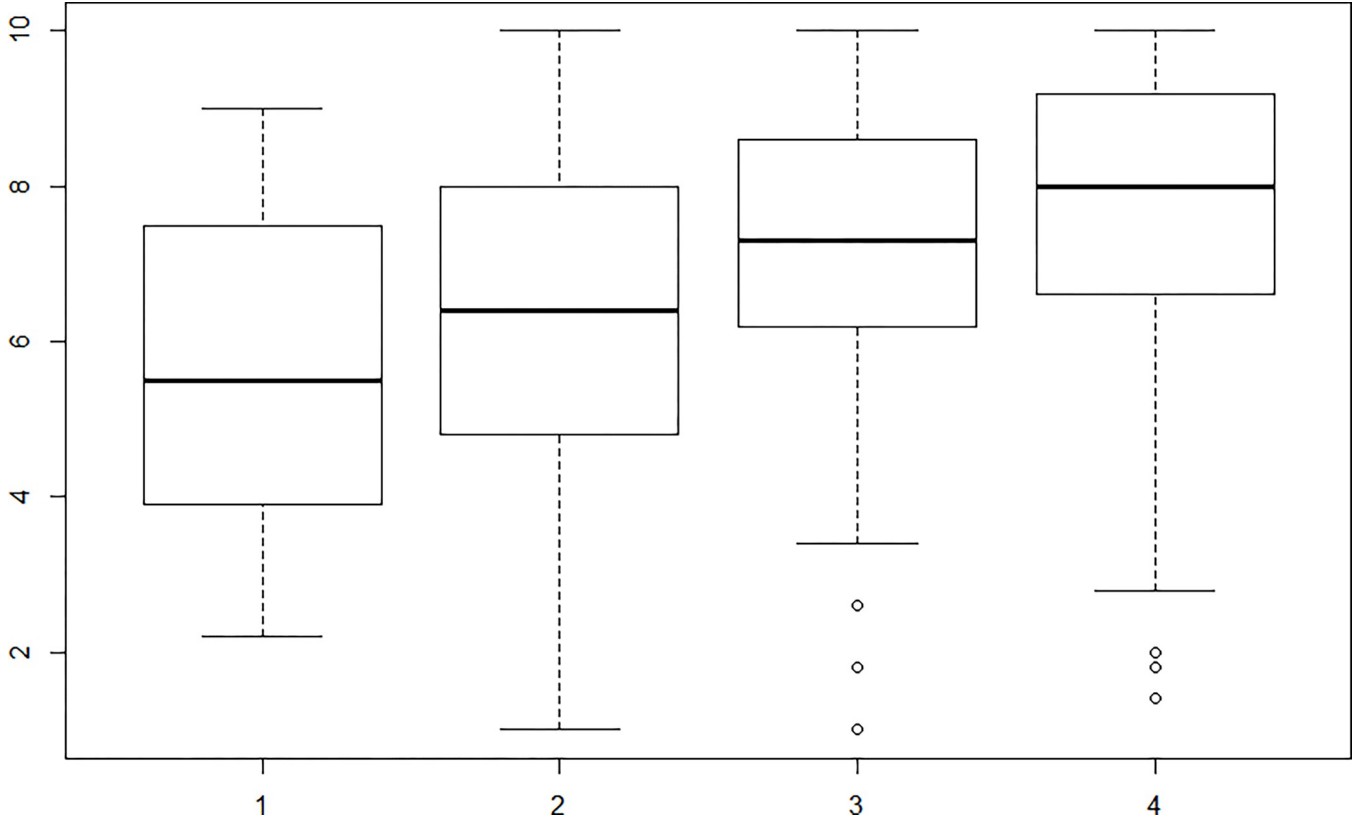

**Fig 2. Relationship factor "External pressure" and the question "If you have tried to lose weight, have you succeeded?".** The scores are equal to the following answers: 1. No, I did not lose weight; 2. Yes, but very little, less than what I expected; 3. Yes, almost what I expected; 4. Yes, what I expected.

In the present study, the instrument was composed of two dimensions: "Physical and emotional discomfort" and "External pressure", in contrast to the Spanish version, which is composed of three dimensions: "Negative emotions", "Social pressure", and "Physical discomfort". In this new distribution, negative emotions were merged with physical discomfort, but item b9 ("I can resist eating when I am in pain") was also eliminated, leaving the scale composed of 11 items. This is also similar to the French validation, which proposed two subscales called "External stimuli" and "Internal stimuli" [17].

This item reorganization may be due to cultural difference in how Chile perceives emotions and physical discomfort. This is particularly true of anxiety, which, albeit a part of everyday human experience, is associated with significant physical symptomatology [27].

The two-factor structure obtained through the exploratory factor analysis is corroborated by the confirmatory analysis, and the elimination of item b9 improved the fit of the scale. The dimension of "Physical and emotional discomfort" includes items that evaluate the ability to resist eating when the person is experiencing negative valence emotional states such as anger, discouragement, and failure, together with physical discomfort. Emerging from the analysis of the association between emotions and eating behavior is the construct of "emotional eating", understood as the tendency to overeat in response to negative emotions and the result of poor interoceptive awareness, in which hunger and satiety signals go unnoticed, resulting in weight gain [28, 29]. The emotional eating factor is composed of the items in the factors of "Negative emotions" and "Physical discomfort" except for Item b1, which was placed in the "External pressure" factor, as noted above.

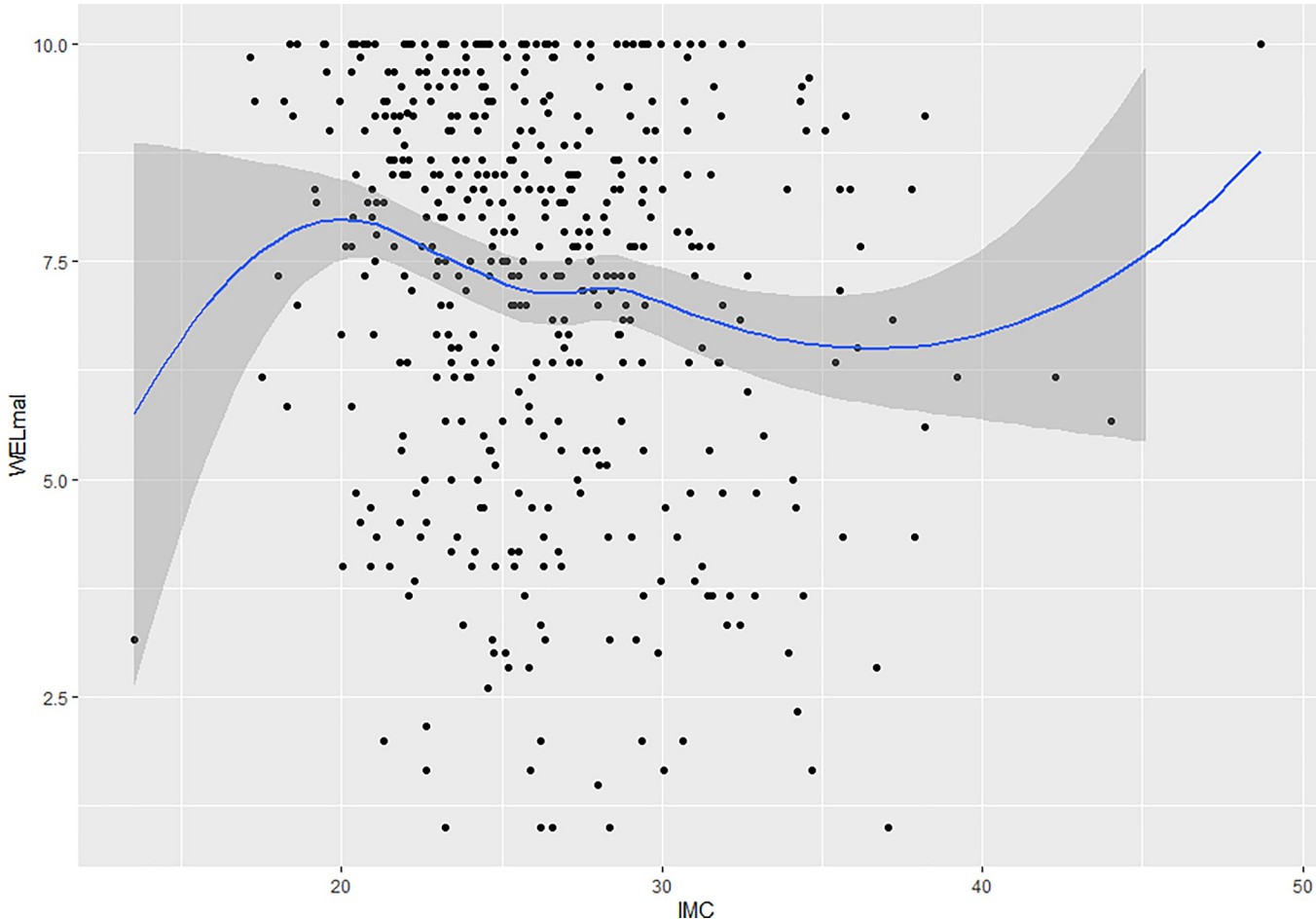

**Fig 3. Relationship between the "Physical and emotional discomfort" factor and the body mass index (BMI).**

The second factor was designated "External pressure" and evaluates the difficulty of resisting eating when there is pressure from the environment. The items that make up this factor are the same as those of the social pressure factor in the Spanish validation [16], to which b1 is added. As previously mentioned, item (b1), "I am able to resist eating when I am anxious (or nervous)", being located in this factor may be because anxiety is usually understood in Chile as being synonymous with overeating (or increased appetite) and not as an emotional state. Another explanation is that anxiety is perceived as an external stimulus that increases appetite, not as an internal emotional state a person can control. For this reason, it was also decided to name the factor "External pressure" and not "Social pressure", as it appears in the Spanish version.

The results of this validation are similar to those of the French validation. Regarding the factor "Physical and emotional discomfort" and its equivalent in the French version for the general population, "Internal stimulus", all of the items coincide except for Item b1 ("I am able to resist eating when I am anxious [or nervous]"), which is placed in this study under the factor "External pressure" and item b9 ("I can resist eating when I am in pain") which was eliminated. In the factor mentioned above and its equivalent in the French version for the general population, "External stimulus", all the factors coincide except one ("I can resist eating even when I am at a party"), which this study did not evaluate as it was eliminated from the Spanish version employed here [16, 17]. This reinforces the proposal of a two-factor structure for the scale.

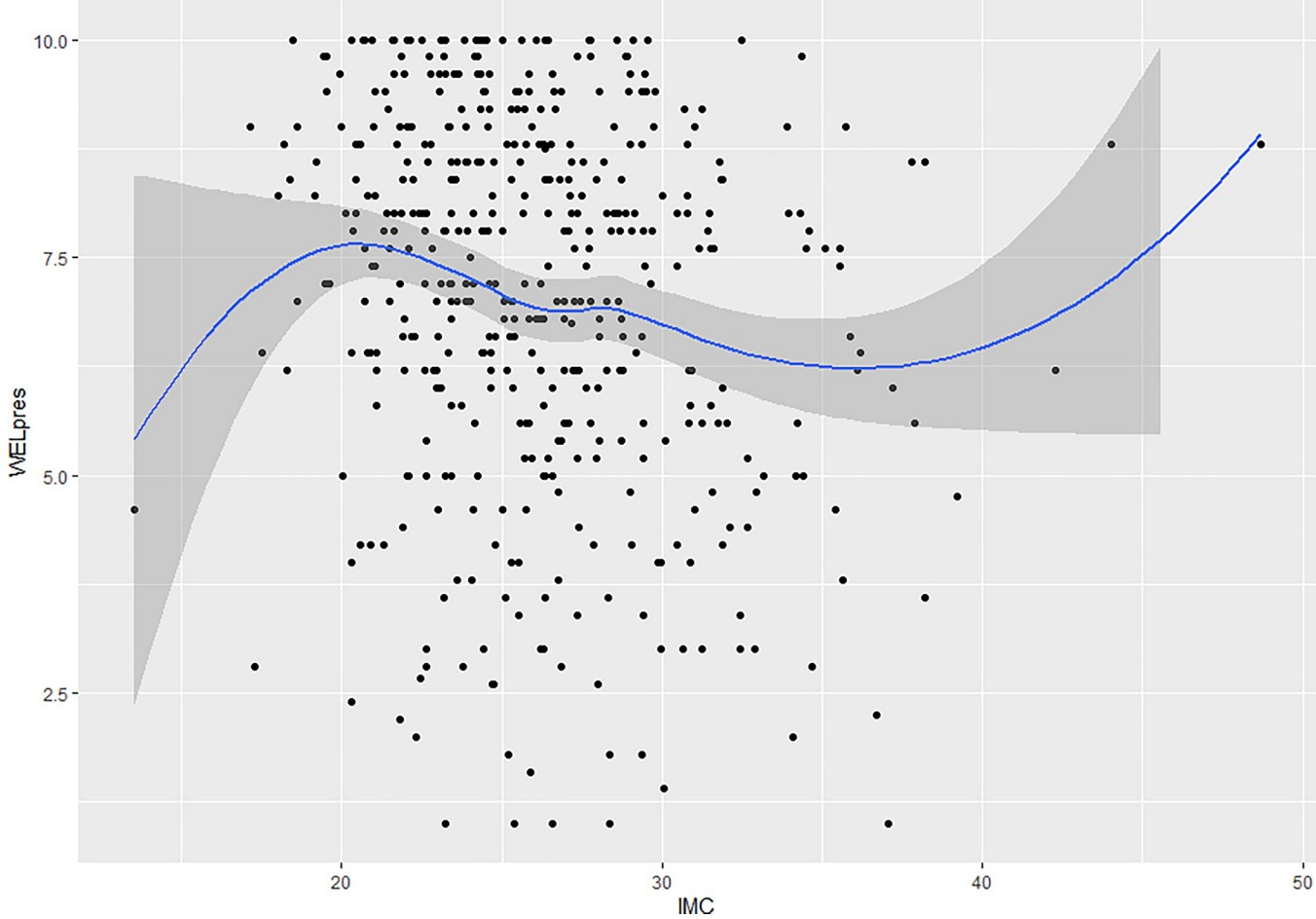

**Fig 4. Relationship between the "External pressure" factor and the body mass index (BMI).**

Regarding the reliability analysis, the instrument shows high internal consistency, with similar results to those reported in other validations [16, 17]. This implies that the scale measures the construct with similar precision. The level of reliability presented by the scale is appropriate for research purposes and, if the rates are available, it could be used for individual assessment.

Reliability is usually estimated with Cronbach's alpha coefficient, which assumes a tau-equivalent measurement model [30]. However, alpha underestimates the true reliability of a measure that is not tau-equivalent, leading to a downward bias [31]. In this light, it is convenient to calculate composite reliability using SEM, which would allow the reliability to be calculated when the loadings or weights of the constructs vary [30, 31]. In this study, both reliability indicators present adequate and close values, which reaffirms an adequate reliability of the scale.

Regarding criterion validity, the WEL correlates inversely as expected with the ESES [15, 16] and the subscales that share similar theoretical content ("Physical and emotional discomfort" in WEL and "Negative affect" in ESES; "External pressure" in WEL and "Socially acceptable circumstances" in ESES). Clark et al. (1991) reported significant relationships between "Socially Acceptable Circumstances" (ESES) and "Availability" (WEL) scales. However, the Spanish authors provided no data on the relationship between "Socially Acceptable

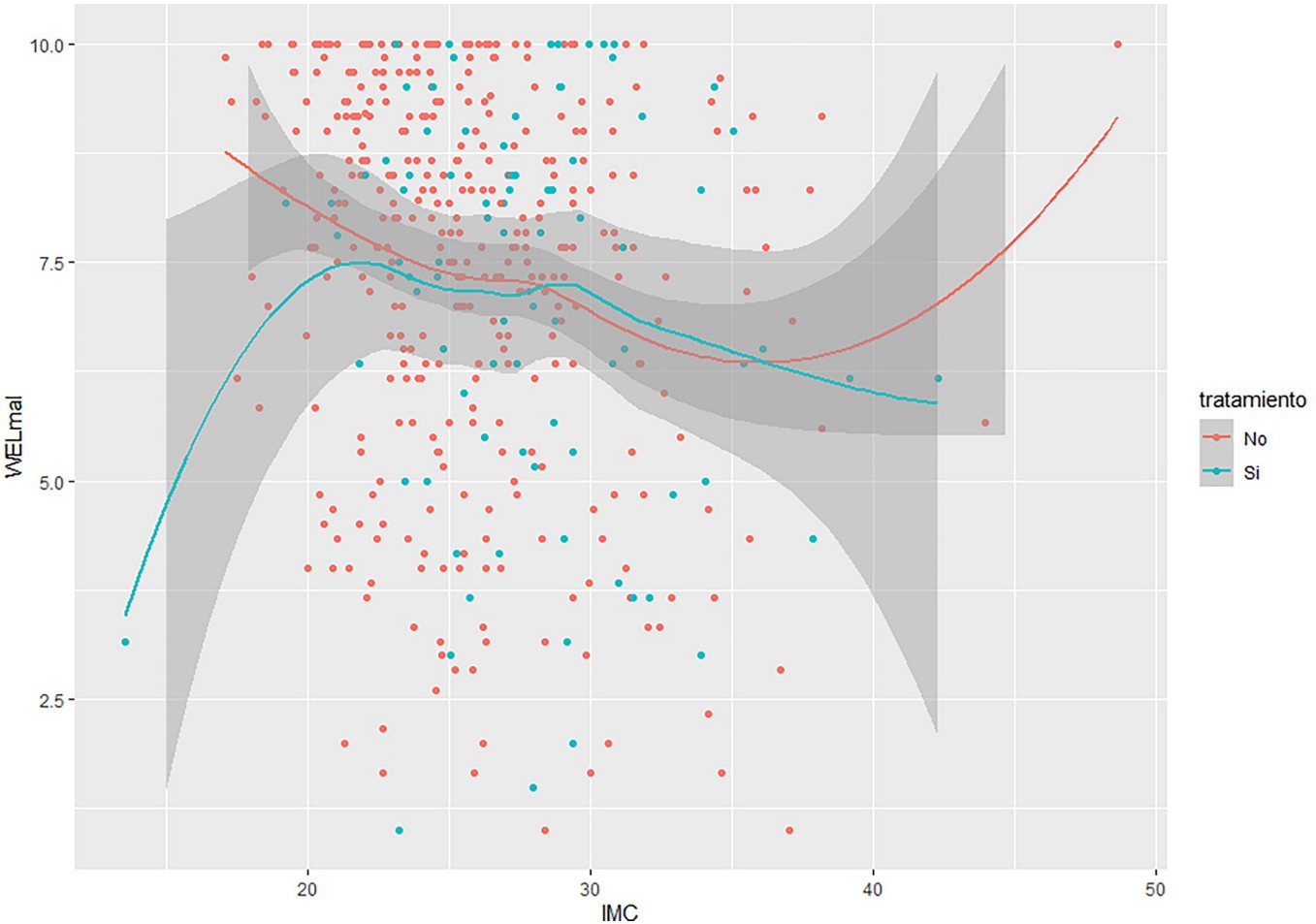

**Fig 5. Relationship between "Physical and emotional discomfort" factor, body mass index, and the attempt to lose weight.**

Circumstances" (ESES) and "Social Pressure" (WEL), which share a greater conceptual affinity. This study presents these data, and, notwithstanding the lower correlation remains adequate. The above contributes further information that the two-factor structure is the best option.

Moreover, the association between WEL and weight loss attempts adds to the evidence that eating self-efficacy is a variable that influences weight loss treatment success [11, 12].

Regarding the relationship between each WEL factor and BMI, there are differences between individuals who have not attempted to lose weight compared to those who have. Individuals who have tried to lose weight and those with low BMIs show low levels of self-efficacy, i.e., they perceive themselves as having little control over their eating-related behaviors, which is characteristic of people who exert excessive control over their food intake, as in anorexia nervosa [32]. On the other hand, high levels of self-efficacy are observed in normal and slightly higher ranges of BMI, which can be associated with successful attempts at weight loss. For people with BMIs in higher ranges, they show low self-efficacy, which would be consistent with failed attempts at weight loss.

This study has some limitations. First, this analysis should be conducted on a larger sample size because although the BMI distribution was similar to those reported by the Chilean National Health Survey [33], a mere 2.3% of the sample was within the malnutrition range, and the same was true of subjects with a BMI over 40 (0.6%).

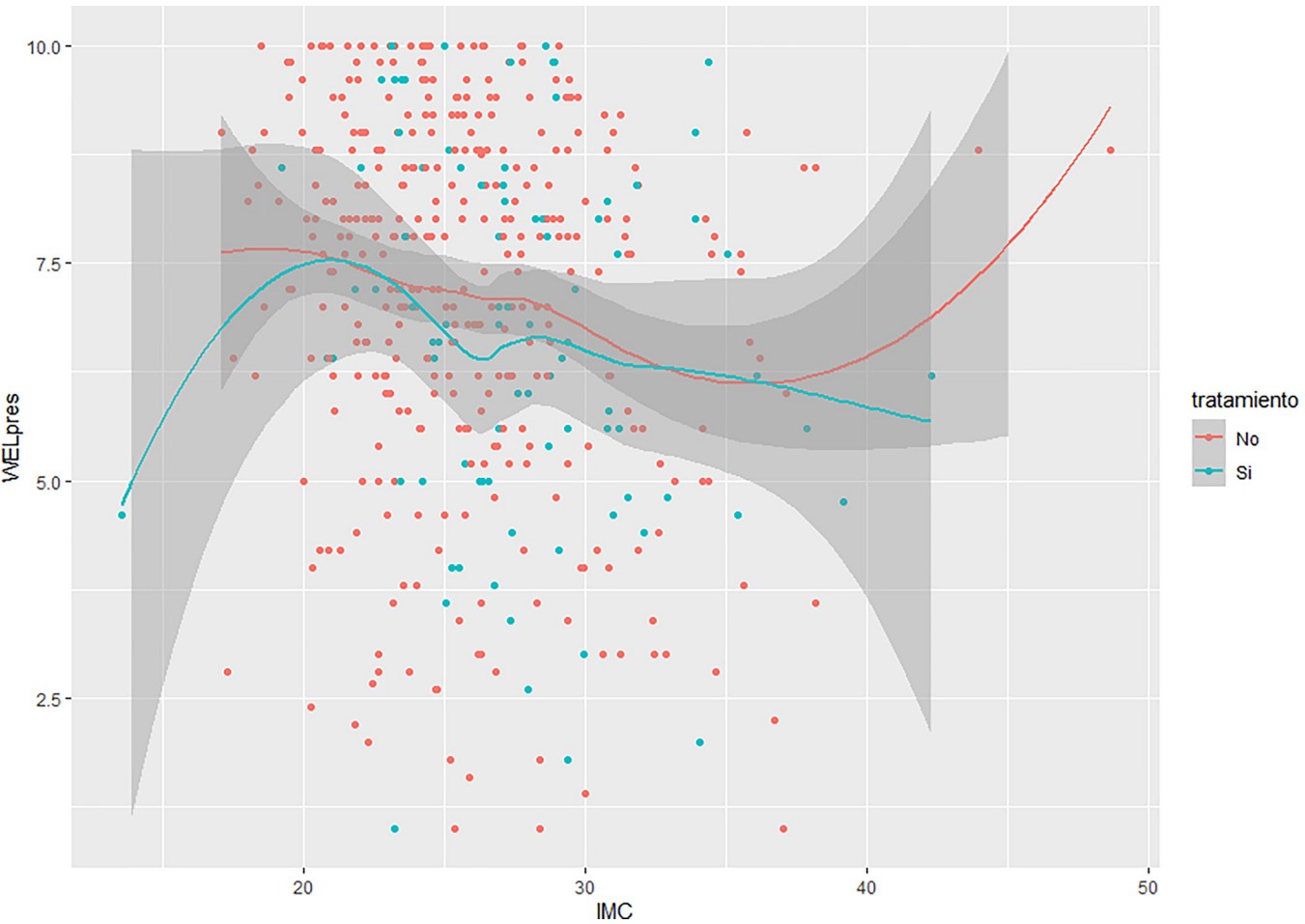

**Fig 6. Relationship between "External pressure" factor, body mass index, and the attempt to lose weight.**

On the other hand, the sample presents certain biases. For example, the greater participation of women may interfere with interpreting the results. Due to the small sample of underweight participants, people with a BMI over 40, and men, the mean comparison tests and the invariance of measurements between BMI categories and between genders should be confirmed with larger general samples.

Another limitation concerns BMI and previous attempts to lose weight as criterion variables. It is known that BMI is not the best method to evaluate the nutritional status of individuals, although the WHO recognizes it as an international standard. It underestimates the total fat mass in people with a lower body mass and may overestimate it in people with a higher body mass [34]. On the other hand, attempts to lose weight are influenced by several factors not considered in this article, so a larger sample with a longitudinal follow-up is needed to demonstrate causality. Exploratory and confirmatory factor analyses were performed with a single sample randomly divided into two groups. Ideally, it would have been advisable to have performed the analysis with different samples and compared them with clinical samples to observe variations in the levels of self-efficacy in people who are trying or have tried to lose weight. The process would affect self-efficacy, but it is not clear whether it is a consequence of the weight loss attempts or prior to them. Another aspect to consider is related to the differences in the level of self-efficacy according to sex. In a recent study in a gender-balanced

sample using the WEL instrument, women showed lower levels of eating self-efficacy [35]. Hence, the above reinforces that since this study has an unequal sample between men and women, the results should be analyzed with caution.

On the other hand, considering the French validation of the instrument, in which the final instrument presents differences in items for the general population and the clinical population [17], this would reinforce the fact that it is necessary to have a clinical sample different from that of the general population.

The fact that invariance was not calculated is a technical limitation of this study. Although calculating invariance would have been beneficial, the gender distribution of the sample is not balanced, and the presence of other covariates has not been taken into account.

Among the strengths of this study are the availability of an instrument to assess eating self-efficacy in the Chilean general population. This study can contribute to evaluating eating behavior in the general population, attending to the psychological factors involved that could influence it. Although the results of this study would indicate differences between the levels of self-efficacy in people with and without obesity, a clinical sample is needed to better assess the differences.

## 5. Conclusion

This study suggests that this version of the WEL with the general population is a valid and reliable instrument to assess eating self-efficacy. Given that it is a psychological factor associated with eating behavior, this might be useful to assess eating self-efficacy in the general population.

With respect to the practical implications, more studies are needed to evaluate the instrument in a clinical population (people with obesity) to ascertain if there are differences and how best to contribute to their treatment.

## Supporting information

**S1 Table. Weight Efficacy Lifestyle Questionnaire: English formulation.**
(DOCX)

**S2 Table. Weight Efficacy Lifestyle Questionnaire: Spanish formulation.**
(DOCX)

## Acknowledgments

The authors thank the participants for collaborating in the study.

## Author Contributions

**Conceptualization:** Mariela Gatica-Saavedra, Gabriela Nazar.

**Formal analysis:** Mariela Gatica-Saavedra, Claudio Bustos.

**Investigation:** Mariela Gatica-Saavedra, Claudio Bustos.

**Methodology:** Mariela Gatica-Saavedra, Gabriela Nazar, Claudio Bustos.

**Writing – original draft:** Mariela Gatica-Saavedra.

**Writing – review & editing:** Gabriela Nazar, Patricia Rubí, Claudio Bustos.

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
