## [Decision Letter · Decision Letter 0]

12 Apr 2023

PONE-D-23-00787Adaptation and validation of the Weight Efficacy Lifestyle Questionnaire (WEL) in a Chilean samplePLOS ONE

Dear Dr. Bustos,

Thank you for submitting your manuscript to PLOS ONE. After careful consideration, we feel that it has merit but does not fully meet PLOS ONE’s publication criteria as it currently stands. Therefore, we invite you to submit a revised version of the manuscript that addresses the points raised during the review process.

I would like to sincerely apologise for the delay you have incurred with your submission. It has been exceptionally difficult to secure reviewers to evaluate your study. We have now received two completed reviews; the comments are available below. The reviewers have raised significant scientific concerns about the study that need to be addressed in a revision.

Please revise the manuscript to address all the reviewer's comments in a point-by-point response in order to ensure it is meeting the journal's publication criteria. Please note that the revised manuscript will need to undergo further review, we thus cannot at this point anticipate the outcome of the evaluation process.

We look forward to receiving your revised manuscript.

Kind regards,

Miquel Vall-llosera Camps

Senior Editor

PLOS ONE

Journal Requirements:

Additional Editor Comments:

PLOS ONE require that studies describing new tools such as new scales/questionnaires or modifications of existing scales/questionnaires meet the additional criteria of utility, validation, and availability. Our author guidelines for studies presenting new methods, software, databases or tools are described in detail at http://journals.plos.org/plosone/s/submission-guidelines#loc-methods-software-databases-and-tools. During your revision of this submission, please ensure you have provided justification for the study, sufficient description of how the new or modified scale/questionnaire was generated, and acceptable measurements of validity and reliability. Please also ensure that the population used to validate the new tool is appropriate and adequately described.

Reviewers' comments:

Reviewer's Responses to Questions

**Comments to the Author**

1. Is the manuscript technically sound, and do the data support the conclusions?

Reviewer #1: Partly

Reviewer #2: Yes

2. Has the statistical analysis been performed appropriately and rigorously? 

Reviewer #1: No

Reviewer #2: Yes

3. Have the authors made all data underlying the findings in their manuscript fully available?

Reviewer #1: No

Reviewer #2: No

4. Is the manuscript presented in an intelligible fashion and written in standard English?

Reviewer #1: No

Reviewer #2: Yes

5. Review Comments to the Author

Reviewer #1: Thanks for submitting this manuscript for review. Overall, I found it an interesting read, with potentially useful applications for public health research experiences and their subsequent actions. I have, however, a series of comments and suggestions that I will list below, following the order proposed by the authors:

- First of all, the paper needs an extensive English writing edition. Please refer to a certified/qualified translator or editor to proofread it.

- The authors state that the objective of their paper is to "analyze the psychometric properties of the Weight Efficacy Lifestyle Questionnaire (WEL) in a sampling of Chilean adults from the general population". Apart from being difficult to understand, the statement does not refer to a validation process. In other words, the authors must be aware that analyzing the psychometric properties of a certain scale in a certain population does not imply validating it. Therefore, a conceptual and literature-based rationale is needed in order to clarify such a categorical error.

- Apart from the aforementioned, the sample is too short for this purpose, and imbalanced in terms of sex/gender, age, and coverage. Although it does not invalidate the study, the methodological issues surrounding this set of facts must be better discussed and formulated in a revised manuscript.

- I feel the criteria used to choose and justify the criterion variables are concerning. While BMI is a good variable for this purpose, it varies over time; secondly, such prior attempts to lose weight are subject to different biases that are basically not controlled nor discussed throughout the paper. Please provide a suitable justification for these technical shortcomings of the study.

- Also, please make the data analysis plan more detailed and sequential in the methods section of the manuscript.

- Line 223: This set of indices (e.g., an RMSEA under .080) does not allow you to determine an "excellent" fit. Instead of using the rules of thumb model, that is rather paracientific, I suggest the authors referring to the March et al. (2004)-based indexes suggested in Table 2 of this paper: https://pubmed.ncbi.nlm.nih.gov/34393528/

- Reliability analysis: Cronbach's alphas are not a strong/robust indicator of it. At best, an indicator of internal consistency. Please calculate and report the CRIs for each factor.

- The complexity and robustness of the analyses and the methods used are sometimes marginal. Why was this small-to-medium-size sample not bootstrapped?

- Most of the study limitations (apart from the aforementioned) are clearly under-acknowledged and under-discussed with methodological sources as support.

- Please kindly translate the figure legends and captions, uniformly.

- Again, please make sure to avoid confusing a psychometrical assessment with a validation. Conclusions are formulated in terms of the first, but it is always confusing when referring to the title of the paper, for which authors need some extra steps.

- Although in view of the few analyses used the first conclusion seems adequate (except for the term "validation", the fact that the authors invite to use it in clinical contexts is totally out of scope (line 448). What are the background or prospects in these regards? Your study does not provide them. Please elaborate and reformulate.

- A suitable database is needed for statistical checks among referees. Could you provide it?

Reviewer #2: TO AUTHORS

The aim of the study was to analyze the psychometric properties of the Weight Efficacy Lifestyle Questionnaire (WEL) in a sample of 469 Chilean adults (69.08% were female, mean age=38,02; SD=10,31). The study is interesting; psychometric analyses were used; the article is well written. Some parts of the manuscript could be clarified. I have made a report and I hope it will be useful to help the authors in the review.

Abstract

- “The sample included 469 people from the general population in Chile…” Revise to: “A total of 469 individuals (69.08% were female, mean age=38,02; SD=10,31) participated in study”.

Background

- The background used to justify the study is robust and in line with the aim.

Material and Methods

- I would like to know how the authors calculated the sample size needed to perform the analyses. Adding this to the first Methods section is important.

- “An instrumental design was used that was geared to the development of psychometric tests and instruments, including their adaptation”. Please provide the exact reference of the instrument designer that was used.

Results

- “The first factor was designated as "Physical and emotional distress… The second factor was termed "External pressure" and includes 5 items…”. Please advise that these labels were created by the authors.

- “A CFA was performed to test the one- and two-factor structure…”. Why did the authors test the one-factor structure?

- “Due to the above, it was decided that a new confirmatory analysis would be performed without including Item b4 in Factor 1”. Why was item b4 excluded from the test? Because of the low factor loading? Why was it not also tested without item b9?

Discussion

- “Regarding the reliability analysis, the instrument shows high internal consistency, with similar results to those reported in other validations (16,17)”. What does that mean? It would be important to explore this result further.

- There are very short paragraphs in this section. For example, the fourth paragraph consists of only one sentence. I suggest reviewing the Discussion to avoid this. Another suggestion is to create a subsection (e.g., Limitations and Strengths) to bring together the last few paragraphs.

Conclusion

- The conclusion is unclear and "inappropriate". It was not the evaluation of the Chilean version of the WEL that showed adequate psychometric properties; in fact, the study found that there was validity and reliability of the Chilean data collected from the WEL. Furthermore, how do the study results clearly support the use of WEL in clinical context if the sample was non-clinical? The fact that there are people with obesity in the sample does not mean that they are " clinical people". I suggest reviewing that section.

6. PLOS authors have the option to publish the peer review history of their article (what does this mean?). If published, this will include your full peer review and any attached files.

Reviewer #1: No

Reviewer #2: **Yes: **Ph.D. Wanderson Roberto da Silva

---

## [Author Response · Author response to Decision Letter 0]

1 Aug 2023

Answers to Reviewers Comments 

Reviewer #1: Thanks for submitting this manuscript for review. Overall, I found it an interesting read, with potentially useful applications for public health research experiences and their subsequent actions. I have, however, a series of comments and suggestions that I will list below, following the order proposed by the authors:

First of all, the paper needs an extensive English writing edition.   Please refer to a certified/qualified translator or editor to proofread it.

Authors’ response: Thank you for your suggestion. The final version of the article was further revised by a certified translator.

The authors state that the objective of their paper is to "analyze   the psychometric properties of the Weight Efficacy Lifestyle   Questionnaire (WEL) in a sampling of Chilean adults from the general  population". Apart from being difficult to understand, the statement  does not refer to a validation process. In other words, the authors must be aware that analyzing the psychometric properties of a certain scale in a certain population does not imply validating it. Therefore, a conceptual and literature-based rationale is needed in order to clarify such a categorical error.

Authors’ response: Thank you for noticing this issue, we agreed with it, so the objective was modified in the Abstract and the main text as follows: this study proposed to validate the WEL in a sample of Chilean adults from the general population

- Apart from the aforementioned, the sample is too short for this purpose, and imbalanced in terms of sex/gender, age, and coverage.  Although it does not invalidate the study, the methodological issues  surrounding this set of facts must be better discussed and formulated

 in a revised manuscript.

Authors’ response: We totally agree with the comment and it is a limitation of the study. The implications of this limitation are discussed in more detail in lines 519 to 526: On the other hand, the sample presents certain biases, for example, the greater participation of women, which may interfere with interpreting the results. And: Another aspect to consider is related to the differences in the level of self-efficacy according to sex. In a recent study in a gender-balanced sample using the WEL instrument, women showed lower levels of eating self-efficacy). The above, therefore, reinforces the fact that since this study has an unequal sample between men and women, the results should be analyzed with caution.

- I feel the criteria used to choose and justify the criterion variables are concerning. While BMI is a good variable for this purpose, it varies over time; secondly, such prior attempts to lose weight are subject to different biases that are basically not controlled nor discussed throughout the paper. Please provide a  suitable justification for these technical shortcomings of the study.

Authors’ response: As mentioned, BMI and previous weight loss are discussed in more detail in lines 501 to 511 and 527 to 535: Another limitation concerns BMI and previous attempts to lose weight as criterion variables. Regarding BMI, it is known that this is not the best method to evaluate the nutritional status of individuals, although it is recognized by the WHO as an international standard, it underestimates the total fat mass in people with lower body mass and may overestimate it in people with higher body mass (32). On the other hand, attempts to lose weight are influenced by several factors that were not considered in this article, so a larger sample with a longitudinal follow-up is needed to demonstrate this relationship.

- Also, please make the data analysis plan more detailed and sequential in the methods section of the manuscript.

Author’s response: As suggested, the data analysis plan was detailed in the Method section.

- Line 223: This set of indices (e.g., an RMSEA under .080) does not   allow you to determine an "excellent" fit. Instead of using the rules  of thumb model, that is rather paracientific, I suggest the authors  referring to the March et al. (2004)-based indexes suggested in Table  2 of this paper: https://pubmed.ncbi.nlm.nih.gov/34393528/

Author’s response: A new analysis was performed that allowed a better fit, closer to what was suggested, with the following results: (X2(43)=77.33, p<0.001; X2/df=1.798; CFI=0.956; TLI=0.943; RMSEA=0.060; SMRM=0.054). Please see lines 318 to 320.

- Reliability analysis: Cronbach's alphas are not a strong/robust  indicator of it. At best, an indicator of internal consistency. Please calculate and report the CRIs for each factor.

Author’s Response: Composite Reliability was calculated for each factor and added to the manuscript. Please see lines 334 to 336.

- The complexity and robustness of the analyses and the methods used  are sometimes marginal. Why was this small-to-medium-size sample not  bootstrapped?

Author’s Response: The confirmatory factor analysis was robust to account for the reliability of the data. No analysis such as anova or t-analysis, was performed with small groups, so the observed differences are assumed reliable by the central limit theorem. Bootstrapped should be evaluated on a case-by-case basis for use in small samples, for example in the case of mediations, and in the case of median calculation, it has been indicated that alternative methods are better to be used. Please see: Zhang M, Liu X, Wang Y, Wang X. Parameter distribution characteristics of material fatigue life using improved bootstrap method. International Journal of Damage Mechanics. 2019;28(5):772-93.

https://journals.sagepub.com/doi/10.1177/1056789518792658

- Most of the study limitations (apart from the aforementioned) are clearly under-acknowledged and under-discussed with methodological  sources as support.

Author’s Response: Thank you for raising this point. As suggested we have strengthened the Limitations of the study. Please see pages 21 to 22.

- Please kindly translate the figure legends and captions, uniformly.

Author’s Response: we have checked and corrected figure legends and captions according to Instructions for Authors. 

- Again, please make sure to avoid confusing a psychometrical  assesment with a validation. Conclusions are formulated in terms of  the first, but it is always confusing when referring to the title of  the paper, for which authors need some extra steps.

Author’s Response: as mentioned above, we agreed with the comment and modified the objective. This new version was in accordance with the title.

- Although in view of the few analyses used the first conclusion seems  adequate (except for the term "validation", the fact that the authors  invite to use it in clinical contexts is totally out of scope (line  448). What are the background or prospects in these regards? Your  study does not provide them. Please elaborate and reformulate.

Author’s Response: the suggestion is accepted. Conclusions were modified considering that this study addressed the general population; thus, self-efficacy can be measured in this group but not in people in treatment to lose weight. Please see lines 576 to 582.

- A suitable database is needed for statistical checks among referees.

  Could you provide it?

Author’s Response: In the Data Availability statement, we have added the link where the database is hosted: https://osf.io/s9uzf/?view_only=3380828e78e44cc790c12714faeb00a4

Reviewer #2: 

The aim of the study was to analyze the psychometric properties of the Weight Efficacy Lifestyle Questionnaire (WEL) in a sample of 469 Chilean adults (69.08% were female, mean age=38,02; SD=10,31). The study is interesting; psychometric analyses were used; the article is well written. Some parts of the manuscript could be clarified. I have

made a report and I hope it will be useful to help the authors in the review.

Abstract

- “The sample included 469 people from the general population in Chile…” Revise to: “A total of 469 individuals (69.08% were female, mean age=38,02; SD=10,31) participated in study”.

Author’s Response: the suggestion is accepted and the modifications were made.

Background

- The background used to justify the study is robust and in line with the aim.

Material and Methods

- I would like to know how the authors calculated the sample size needed to perform the analyses. Adding this to the first Methods section is important.

Author’s Response: The suggestion is accepted and the modifications were made. You can find this information on lines 162 to 166.

- “An instrumental design was used that was geared to the development of psychometric tests and instruments, including their adaptation”.

Please provide the exact reference of the instrument designer that was used.

Author’s Response: We agree that was not clear enough the study design, which in this case refers to an instrumental design. To clarify the above, the reference in which the above is specified is added on lines155-159-.

Results

- “The first factor was designated as "Physical and emotional distress…The second factor was termed "External pressure" and includes 5 items…”. Please advise that these labels were created by the authors.

Author’s Response: the suggestion is accepted and the modifications were added. You can find this information on lines 304 to 309.

- “A CFA was performed to test the one- and two-factor structure…”. Why did the authors test the one-factor structure?

Author’s Response: We test the one-factor structure because in the analysis with Velicer's MAP we obtained a minimum of 0.03 with 1 factor, the same as in Horn's Parallel Analysis. The aforementioned was indicated in lines 289 to294.

- “Due to the above, it was decided that a new confirmatory analysis would be performed without including Item b4 in Factor 1”. Why was item b4 excluded from the test? Because of the low factor loading? Why was it not also tested without item b9?

Author’s Response: Thank you for your suggestion. New analyses are performed, one excluding item b4 and another one excluding item b9. The best fit is obtained without item b9, for this reason, it is decided to perform all the analyses again without that item. The details of the aforementioned can be found between lines 322 and 342.

Discussion

- “Regarding the reliability analysis, the instrument shows high internal consistency, with similar results to those reported in other validations (16,17)”. What does that mean? It would be important to explore this result further.

Author’s Response: Details of your suggestion are provided between lines 480 and 484

- There are very short paragraphs in this section. For example, the fourth paragraph consists of only one sentence. I suggest reviewing the Discussion to avoid this. Another suggestion is to create a subsection (e.g., Limitations and Strengths) to bring together the last few

paragraphs.

Author’s Response: the suggestion was accepted and the modifications were added. Please see Limitations which were discussed in more detail in lines 519 to 526:

Conclusion

- The conclusion is unclear and "inappropriate". It was not the evaluation of the Chilean version of the WEL that showed adequate psychometric properties; in fact, the study found that there was validity and reliability of the Chilean data collected from the WEL.

Author’s Response:

Furthermore, how do the study results clearly support the use of WEL in clinical context if the sample was non-clinical? The fact that there are people with obesity in the sample does not mean that they are" clinical people". I suggest reviewing that section.

Author’s Response the suggestion is accepted and the modifications were made in the Conclusion emphasizing that the sample was a general population thus, no conclusions can be made about any other particular groups.

---

## [Decision Letter · Decision Letter 1]

31 Aug 2023

PONE-D-23-00787R1Adaptation and validation of the Weight Efficacy Lifestyle Questionnaire (WEL) in a Chilean samplePLOS ONE

Dear Dr. Bustos,

Thank you for submitting your manuscript to PLOS ONE. After careful consideration, we feel that it has merit but does not fully meet PLOS ONE’s publication criteria as it currently stands. Therefore, we invite you to submit a revised version of the manuscript that addresses the points raised during the review process.

We look forward to receiving your revised manuscript.

Kind regards,

Mohammad Asghari Jafarabadi

Academic Editor

PLOS ONE

Journal Requirements:

Please review your reference list to ensure that it is complete and correct. If you have cited papers that have been retracted, please include the rationale for doing so in the manuscript text, or remove these references and replace them with relevant current references. Any changes to the reference list should be mentioned in the rebuttal letter that accompanies your revised manuscript. If you need to cite a retracted article, indicate the article’s retracted status in the References list and also include a citation and full 

Reviewers' comments:

Reviewer's Responses to Questions

**Comments to the Author**

1. If the authors have adequately addressed your comments raised in a previous round of review and you feel that this manuscript is now acceptable for publication, you may indicate that here to bypass the “Comments to the Author” section, enter your conflict of interest statement in the “Confidential to Editor” section, and submit your "Accept" recommendation.

Reviewer #1: (No Response)

Reviewer #2: (No Response)

2. Is the manuscript technically sound, and do the data support the conclusions?

Reviewer #1: Partly

Reviewer #2: Yes

3. Has the statistical analysis been performed appropriately and rigorously? 

Reviewer #1: Yes

Reviewer #2: Yes

4. Have the authors made all data underlying the findings in their manuscript fully available?

Reviewer #1: Yes

Reviewer #2: Yes

5. Is the manuscript presented in an intelligible fashion and written in standard English?

Reviewer #1: Yes

Reviewer #2: Yes

6. Review Comments to the Author

Reviewer #1: Thanks for your revisions and amendments. I have some further comments, which should be addressed by the authors for their resubmission:

- I have been cross-checking the survey data, finding that there are some potential bias sources that, although not under authors' control, must be addressed in the manuscript.

- Please add a suitable elaboration on how missing data were handled, and what kind of data were mainly omitted. Also, please discuss this in the light of both the current study settings and any relevant technical source contained in literature.

- Also, did you assess invariance? I guess there are probably sex and other categorical factor-based invariance issues. In case you tested it, please add it to the paper (briefly). If not, you can add this as a technical limitation of the study.

- Thanks for adding the composite reliability analysis outcomes. It improves a lot the assumptions related to reliability of your paper. Please also make this evident (and literature-supported) in the discussion of the manuscript.

- In the appendix, please add another column with the most suitable translation of the items to the English (given that most of your readers are supposed to be non-Spanish but English speakers).

- In conclusions section, the sentence "More studies are needed to (...)" is not a conclusion of your study (you did not address it from an empirical perspective). Therefore, please raise it as a practical implication, in a subsequent paragraph.

- Finally, and as a matter of courtesy with your reviewers, polease improve the way of track changes in a revised manuscript version. Using strikethrough text makes it very difficult to follow along. Better use the highlight tool, and/or a different color font.

Reviewer #2: The authors reported that they agreed with all my comments and revised the manuscript; however, I have not been able to find the changes in the manuscript. The number of lines mentioned by the authors in the response letter does not correspond to the numbers in the manuscript. In order for me to properly proofread the manuscript, authors need to highlight changes to the text (eg, using yellow color).

7. PLOS authors have the option to publish the peer review history of their article (what does this mean?). If published, this will include your full peer review and any attached files.

Reviewer #1: No

Reviewer #2: No

---

## [Author Response · Author response to Decision Letter 1]

3 Oct 2023

October 3, 2023

Answers to Reviewers' Comments 

Reviewer #1: Thanks for your revisions and amendments. I have some further comments, which should be addressed by the authors for their resubmission:

I have been cross-checking the survey data, finding that there are some potential bias sources that, although not under authors' control, must be addressed in the manuscript.

- Please add a suitable elaboration on how missing data were handled, and what kind of data were mainly omitted. Also, please discuss this > in the light of both the current study settings and any relevant technical source contained in literature.

Authors answer: No data imputation technique was necessary because the R program only performed the analyses with complete data. This information is found on lines 225 and 226 of the manuscript with track changes.

- Also, did you assess invariance? I guess there are probably sex and other categorical factor-based invariance issues. In case you tested > it, please add it to the paper (briefly). If not, you can add this as a technical limitation of the study.

Although it would have been good to calculate invariance, the sample is unbalanced between men and women, and the presence of other covariates has not been controlled. This will be reported as a limitation of the study and you can find it on lines 476 to 479 of the manuscript with track changes.

 - Thanks for adding the composite reliability analysis outcomes. It improves a lot the assumptions related to reliability of your paper. Please also make this evident (and literature-supported) in the discussion of the manuscript.

The reviewer's suggestion was accepted and the implications of the composite reliability results were included in the Discussion. This information is found on lines 417 to 423 of the manuscript with track changes.

- In the appendix, please add another column with the most suitable translation of the items to the English (given that most of your readers are supposed to be non-Spanish but English speakers).

We appreciate the reviewer's suggestion and have added the English translation of the scale. This information is found on lines 628 to 629 of the manuscript with track changes.

- In conclusions section, the sentence "More studies are needed to (...)" is not a conclusion of your study (you did not address it from an empirical perspective). Therefore, please raise it as a practical implication, in a subsequent paragraph.

We appreciate the reviewer's suggestion and incorporate it into the manuscript. This information is found on line 493 of the manuscript with track changes.

- Finally, and as a matter of courtesy with your reviewers, please improve the way of track changes in a revised manuscript version. Using strikethrough text makes it very difficult to follow along. Better use the highlight tool, and/or a different color font.

We apologize for the confusing nature of the above manuscript. Changes are highlighted in yellow and deleted in red.

Reviewer #2: The authors reported that they agreed with all my comments and revised the manuscript; however, I have not been able to find the changes in the manuscript. The number of lines mentioned by the authors in the response letter does not correspond to the numbers in the manuscript. In order for me to properly proofread the manuscript, authors need to highlight changes to the text (eg, using yellow color). 

We apologize for the confusing nature of the above manuscript. Changes are highlighted in yellow and deleted in red.

---

## [Decision Letter · Decision Letter 2]

18 Oct 2023

Adaptation and validation of the Weight Efficacy Lifestyle Questionnaire (WEL) in a Chilean sample

PONE-D-23-00787R2

Dear Dr. Bustos,

We’re pleased to inform you that your manuscript has been judged scientifically suitable for publication and will be formally accepted for publication once it meets all outstanding technical requirements.

Kind regards,

Mohammad Asghari Jafarabadi

Academic Editor

PLOS ONE

Reviewers' comments:

Reviewer's Responses to Questions

**Comments to the Author**

1. If the authors have adequately addressed your comments raised in a previous round of review and you feel that this manuscript is now acceptable for publication, you may indicate that here to bypass the “Comments to the Author” section, enter your conflict of interest statement in the “Confidential to Editor” section, and submit your "Accept" recommendation.

Reviewer #1: All comments have been addressed

Reviewer #2: All comments have been addressed

2. Is the manuscript technically sound, and do the data support the conclusions?

Reviewer #1: Yes

Reviewer #2: (No Response)

3. Has the statistical analysis been performed appropriately and rigorously? 

Reviewer #1: Yes

Reviewer #2: Yes

4. Have the authors made all data underlying the findings in their manuscript fully available?

Reviewer #1: Yes

Reviewer #2: Yes

5. Is the manuscript presented in an intelligible fashion and written in standard English?

Reviewer #1: Yes

Reviewer #2: Yes

6. Review Comments to the Author

Reviewer #1: For this version of their paper, the authors have done an outstanding job.

Reading the paper for a second time, I found how most of my comments and suggestions were adequately considered and incorporated by the authors. The responses are overall sound, and the rationales supporting the key points expressed in the manuscript are now quite clearer.

As a pending matter, I still find the introduction somewhat improvable, and the methods not totally strong (especially as for the psychometric analyses, still basic), but I guess the information already appended in both the literature review and the analyses made does reach an acceptable minimum.

Finally, and regarding my suggestion to the editor, I believe the paper could be considered as publishable as it is. Good job and thanks for all your efforts.

Best wishes.

Reviewer #2: The authors successfully revised the manuscript following my comments. The revised manuscript is clearer and more scientifically adequate.

7. PLOS authors have the option to publish the peer review history of their article (what does this mean?). If published, this will include your full peer review and any attached files.

Reviewer #1: No

Reviewer #2: **Yes: **Ph.D. Wanderson Roberto da Silva

---

## [Editor Report · Acceptance letter]

23 Jan 2024

PONE-D-23-00787R2 

PLOS ONE

Dear Dr. Bustos, 

I'm pleased to inform you that your manuscript has been deemed suitable for publication in PLOS ONE. Congratulations! Your manuscript is now being handed over to our production team.

Kind regards, 

on behalf of

Professor Mohammad Asghari Jafarabadi 

Academic Editor

PLOS ONE